# New Insights into Aptamers: An Alternative to Antibodies in the Detection of Molecular Biomarkers

**DOI:** 10.3390/ijms25136833

**Published:** 2024-06-21

**Authors:** Michaela Domsicova, Jana Korcekova, Alexandra Poturnayova, Albert Breier

**Affiliations:** 1Centre of Biosciences, Institute of Molecular Physiology and Genetics, Slovak Academy of Sciences, Dúbravská Cesta 9, 84005 Bratislava, Slovakia; michaela.domsicova@savba.sk (M.D.); jana.korcekova@savba.sk (J.K.); alexandra.poturnayova@savba.sk (A.P.); 2Institute of Biochemistry and Microbiology, Faculty of Chemical and Food Technology, Slovak University of Technology in Bratislava, Radlinského 9, 81237 Bratislava, Slovakia

**Keywords:** aptamer, SELEX, ELISA, ELASA, QCM aptasensor, cancer detection

## Abstract

Aptamers are short oligonucleotides with single-stranded regions or peptides that recently started to transform the field of diagnostics. Their unique ability to bind to specific target molecules with high affinity and specificity is at least comparable to many traditional biorecognition elements. Aptamers are synthetically produced, with a compact size that facilitates deeper tissue penetration and improved cellular targeting. Furthermore, they can be easily modified with various labels or functional groups, tailoring them for diverse applications. Even more uniquely, aptamers can be regenerated after use, making aptasensors a cost-effective and sustainable alternative compared to disposable biosensors. This review delves into the inherent properties of aptamers that make them advantageous in established diagnostic methods. Furthermore, we will examine some of the limitations of aptamers, such as the need to engage in bioinformatics procedures in order to understand the relationship between the structure of the aptamer and its binding abilities. The objective is to develop a targeted design for specific targets. We analyse the process of aptamer selection and design by exploring the current landscape of aptamer utilisation across various industries. Here, we illuminate the potential advantages and applications of aptamers in a range of diagnostic techniques, with a specific focus on quartz crystal microbalance (QCM) aptasensors and their integration into the well-established ELISA method. This review serves as a comprehensive resource, summarising the latest knowledge and applications of aptamers, particularly highlighting their potential to revolutionise diagnostic approaches.

## 1. Introduction

A new class of oligonucleotide-based molecular recognition elements—aptamers—has emerged as complementary agents that extend the capabilities of diagnostic methods based on antibodies and other biorecognition proteins [1].

Although it is not yet possible to unambiguously design the structure of aptamers with affinity for specific molecules, the SELEX system (Systematic Evolution of Ligands by Exponential Enrichment) is such a method that allows for the generation of aptamers with high affinity and specificity for a wide range of target ligands, such as proteins, peptides, glycosides, nucleotides, amino acids, antibiotics, and small organic or inorganic molecules, but can even be applied to whole cells and tissues [2]. These make aptamers attractive for developing new alternative approaches in traditional diagnostic, therapeutic, or biorecognition methods.

## 2. Aptamers

### 2.1. Aptamer Structure

Aptamers with a folded three-dimensional structure are typically 20 to 100 mononucleotide-sized DNA or RNA molecules containing double-stranded portions and single-stranded loops within the molecule. These loops form fine spatial motifs capable of high-affinity and specific binding to a concatemeric ligand. The complementary regions in the oligonucleotide molecule facilitate the formation of a defined, energetically favourable 3D structure through hydrogen bonding, which enables ligand binding (Figure 1A) [3]. Nevertheless, alterations in intramolecular base pairing, hydrogen bonding, and other van der Waals interactions, such as hydrophobic interactions, can result in the formation of more complex tertiary structures. These structures exhibit a sequence-dependent three-dimensional conformation, which includes loops, quadruplexes, pseudoknots, bulges, and hairpins (Figure 1B–E). This enables the specific recognition and binding to a target. The binding of aptamers to their targets can result in a range of biochemical effects, including the destruction, antagonism, agonism, suppression, or inhibition of the target [4].

Aptamers recognise molecules through intermolecular interactions of a van der Waals nature, such as the joining of aromatic rings and electrostatic and/or hydrogen bonds with the target ligand. In some cases, the ligand binding domain of the unbound aptamer remains completely unstructured, and the specific tertiary structure is only achieved after ligand binding. Aptamers bind their ligands by folding and encapsulating using an ‘induced fit mechanism’. The final bioactive conformation is achieved only during the binding process, as the interaction between the ligand and the aptamer induces the formation of new secondary and tertiary structure elements [4,5,6].

Aptamers can also form a double helix structure by hybridising with their complementary sequences based on their nucleic acid origin. Then, the competitive interaction between aptamers and target ligands leads to the disappearance of the double helix, resulting in a single-stranded structure [3].

### 2.2. Advantages of Aptamers

Once the aptamer sequence is identified, it can be reproduced easily, resulting in a preparation with high purity and consistent performance. Another advantage is how easily they can be modified with various labels or markers, allowing for an easy detection of target molecules bound to aptamers. Common labels include fluorescent probes and electrochemical indicators that cause a measurable electric signal [7].

Perhaps the most surprising feature of aptamers is their reusability. Unlike some biological tools that wear out after a single use, aptamers can be regenerated after binding to their target. This regeneration process is often simple and does not affect the aptamer’s ability to recognize its target with high affinity and selectivity, making them a sustainable and cost-effective molecular recognition tool. However, it is important to note that with the immobilisation of the aptamer on a sensor surface, or significant changes in the surrounding environment, the aptamer structure can be slightly altered. Such alterations can consequently affect its interaction with the target molecule.

Aptamers also possess the remarkable ability to discriminate between different molecules, including their chirality, and to recognise the folded shapes (3D structure) of proteins [8]. Nucleic acids, particularly RNA, can be cleaved by nucleases, which are ubiquitous in nature. Consequently, the risk of degradation by ubiquitous RNases and DNases must be overcome, necessitating the protection of aptamer structure from degradation and the assurance of their long-term functionality [9].

The dissociation constant (K_d_) of aptamers, expressing the inverse ratio of the aptamer’s affinity to a specific ligand, ranges from micromolarity to picomolarity. This affinity is often comparable to or better than the affinity of monoclonal antibodies to their antigens [10].

The ability of aptamers to specifically recognize targets from their derivatives has been demonstrated in various articles, such as the theophylline-binding aptamer, which exhibited a 10,000-fold higher binding capacity compared to caffeine (the only difference being the methyl group) [11]. The L-arginine-binding aptamer showed 12,000-fold multiple affinity compared to D-arginine [12], and the aptamer binding oxytetracycline recognised it specifically instead of tetracycline (the only difference being the OH group) [13].

In addition to high affinity and specificity, aptamers have several more advantages, allowing their wide applications. They are easy to work with due to their thermostability and can be easily regenerated within minutes after denaturation because they undergo reversible denaturation. This flexibility of structure is useful in the development of new types of sensing methods with multiple uses [3]. Aptamers can be readily modified and linked to labelled molecules or immobilised on the surface of beads or carriers for various applications [14]. This advantage is particularly significant in the diagnosis and application of biosensors, where uniform arrangement and immobilisation in analytical systems are crucial. Labelling aptamers with signal-generating molecules is a common method for signal production or amplification in biosensors [15].

Their possible use in the clinical field is further supported by the fact that aptamers which are administered in an in vivo system produce no or only a rare immune response [15]. Their small size (generally less than 20 kDa) facilitates the easy monitoring of sample bioavailability and makes their penetration into cells and their transfer or immunisation in any medium easier to achieve, similar to liposomes [16].

Nucleic acid aptamers exhibit enhanced stability and tolerance towards chemical modification, including fluorescent dyes, radionuclides, redox labels, and nanomaterial conjugation. They also demonstrate resilience in harsh environmental conditions, such as changes in temperature, pH, humidity, and the presence of organic solvents [17]. Aptamers can be easily amplified by PCR and can be cloned to appropriate vectors and expressed inside different cells [15,18]. The fact that aptamers do not require post-translational chemical modifications allows their production in a variety of prokaryotes, ensuring simple and inexpensive processes.

One of the most significant advantages of using aptamers is that they do not require animals for selection and production. Aptamers can be isolated and chemically synthesised in vitro without the need for in vivo immunisation in high purity at a low cost and without batch-to-batch differences, making them easily accessible [8]. This is advantageous because producing antibodies against certain substrates, such as proteins that are structurally similar to endogenous proteins and toxic compounds, can be difficult. Over the years, the SELEX process of aptamer generation has been modified to develop aptamers with high affinity and selectivity in the most efficient and automated manner possible [19]. Using these advantages has led to aptamers being successfully applied as recognition elements in medical diagnostics and biosensor development [20]. The primary limitation to the extensive utilisation of aptamers in practice is the lack of clarity regarding the relationship between the aptamer’s structural characteristics and its binding capabilities, in contrast to the well-established knowledge of antibody-target interactions, which are widely used in practical procedures. This limitation leads to the fact that it is not possible to replace the empirical selection of aptamers with a targeted procedure of their tailored design for the target ligand. It is therefore necessary to focus on bioinformatic approaches that would help to clarify this relationship, so that it is possible to optimise the structure of the aptamer in a targeted manner for the target ligand.

## 3. Generating Aptamers—SELEX System

Physico-chemical binding parameters, such as affinity, kinetics, and thermodynamics, are crucial in the development and implementation of aptamers. The K_d_ constant indicates the binding affinity between the aptamer and ligand, with a lower value indicating higher binding affinity [21]. To select relevant candidates during aptamer development, an affinity measure is often used to rank the set of aptamers. Furthermore, affinity enables the evaluation of aptamer selectivity towards a single ligand or the specificity towards multiple ligands. Binding kinetics refer to the time-dependent dynamic component of the binding force between the aptamer and ligand. The association rate constant (k_a_) describes the association of the aptamer and ligand to form a binary or higher order complex over time. The dissociation rate constant K_d_ describes the rate at which the aptamer and ligand dissociate and is therefore a measure of the stability of the aptamer–ligand complex over time. To ensure the proper functionality of the aptamer in its final application, such as in diagnostics, it is important to select aptamers with the desired binding kinetics during the development phase [22].

In 1990, three laboratories independently developed a technique for isolating functional oligonucleotides from a randomly synthesised library of nucleic acids composed of over 10^5^ different sequences, demonstrating affinity for the target [23,24,25].

The SELEX method (Figure 2) is used for the rapid and reliable in vitro selection of aptamers for binding to a specific structure. It is an aptamer screening process that is influenced by several parameters, such as the properties of the target ligand, the design of the random DNA library, the selection conditions, and the efficiency of the separation methods. The selection process is repeated six to ten times with an increasing stringency of aptamer binding conditions. This results in affinity values that are comparable to the affinity of recognition molecules of different origin. The SELEX process involves four basic steps: binding, selection, amplification, and separation [26].

### 3.1. Steps of the SELEX Process

#### 3.1.1. DNA Library Design

The SELEX process starts with the chemical synthesis of a single-stranded (ssDNA) library consisting of random sequences. The library is flanked in the middle by defined binding sites for primers at the 5′ and 3′ ends, with each individual ssDNA having different sequences. The diversity of the library is dependent on the length of the random region. For a length of n nucleotides, 4n different sequences are generated [27].

Typically, an initial single-stranded DNA (ssDNA) sample composed of approximately 10^15^ different sequences is used. This allows for the generation of ligand-specific sequences with a high probability, corresponding to a length of approximately 25 nucleotides. It is noteworthy that only a short section of the entire aptamer is necessary for binding to the target molecule [28]. This implies that even a small library can be used for successful aptamer screening, which is cost-effective and easy to handle. However, longer random sequences in the library are more suitable for providing high structural complexity, which is important in isolating high-affinity aptamers. Therefore, a longer library may increase the chance of successful aptamer selection [29].

The constant region’s design, which is the binding site of the aptamer, should be appropriately designed to prevent non-specific pairing during PCR amplification and reduce the possibility of base pairing between the two constant regions. It is crucial to design the constant region correctly because the DNA sequence should undergo multiple repeat amplifications. Otherwise, any undesired nucleotides after hundreds of PCR cycles could be amplified in the final pool of aptamers [30].

#### 3.1.2. Characterisation of Aptamer–Target Ligand Binding

In the next step, the initial random ssDNA library is incubated with the target ligand. In the case of selecting RNA aptamers, the ssDNA library must first be transformed into an RNA library before incubation. RNA libraries were commonly used for aptamer selection at the beginning of aptamer research because RNA folds better into 3D complex structures, resulting in stronger molecular interactions with the target ligand. RNA aptamers are often considered superior to DNA aptamers in terms of their affinity and specificity due to the 2-hydroxyl group present on ribose in RNA that can facilitate the formation of additional hydrogen bonds between the aptamer and the target ligand. However, for in vivo applications such as therapeutics, molecular imaging, or drug distribution, both RNA and DNA aptamers need to be modified to resist degradation by nucleases, which can be expensive. RNA aptamers are less stable than DNA aptamers due to the presence of a wider variety of RNases than DNases in the environment. Therefore, for in vitro applications such as biosensor development, DNA aptamers do not require modification to improve stability, whereas RNA aptamers still need to be modified due to their low stability. Although ssDNA folds into a 3D configuration, its folded structure is less stable than the corresponding RNA sequence. DNA aptamers are therefore less expensive to work with than RNA aptamers [29,31].

#### 3.1.3. Selection of Suitable Aptamers

Selection is the most critical step in the SELEX process. In this step, a random DNA or RNA library is incubated with a ligand. The nucleic acid–ligand complex is subsequently separated from free or weakly bound nucleotides [26].

This step is crucial for isolating aptamers with high affinity and high specificity from a highly diverse oligonucleotide library. During incubation, target molecules interact with nucleic acids in free form or as immobilised on the ligand [32].

#### 3.1.4. Amplification of Selected Aptamers

The oligonucleotides bound to the target ligand are amplified through reverse transcription PCR for an RNA library or PCR for a DNA library, generating new oligonucleotides for the next SELEX step. It is important to optimise PCR conditions, such as the number of cycles, depending on the primer and library design, as PCR efficiency in the SELEX process is not high due to random sequence regions in the nucleotide sample [33].

#### 3.1.5. Separation of Amplified Aptamers

Following PCR amplification, the enriched oligonucleotide sample exists as double-stranded DNA. The dsDNA is then separated into individual ssDNA strands, which are subsequently incubated with the target ligand during the next step of SELEX. To select RNA aptamers, it is necessary to transcribe the ssDNA into RNA. The streptavidin–biotin system is commonly employed for aptamer selection. This method involves incorporating a biotin molecule into unwanted helices during PCR amplification using a biotinylated reverse primer. The resulting biotin-labelled dsDNA (reverse 3′–5′ helices) is then incubated with or passed through a column containing streptavidin-coated beads. The 5′–3′ DNA originating from nucleic acids bound to the target molecule is separated by alkaline denaturation or affinity purification, while the biotinylated reverse structures are captured on streptavidin carriers [33,34,35,36].

The fixation of the target ligand on a solid support allows for an easy separation of the nucleic acids that are bound to the target ligand from those that are unbound or weakly bound. This method is highly effective for low mass target molecules. However, ligand immobilisation can cause conformational changes and interfere with library binding [37]. It is also important to consider the non-specific interactions of the nucleic acid with the solid ligand or linker molecules.

To avoid the issue of the difficult elution of nucleic acids strongly bound to the target, aptamers with K_d_ values in the nano- to picomolar range use the SELEX process with free-form target molecules. This method is primarily employed for macromolecular target compounds due to the difficulty in separating low-molecular-weight complexes of the target compound and nucleotides in free form. However, the efficiency of separation is typically low [38].

SELEX processes traditionally use filtration through a nitrocellulose membrane or affinity chromatography with a column containing target compounds immobilised on beads for separation [23]. The filtration method removes oligonucleotides not bound to the target ligands from the aptamer–ligand complexes. Affinity chromatography separates oligonucleotides interacting with a target molecule from a nucleic acid sample using a column packed with a target molecule immobilised on beads. Affinity chromatography often requires multiple SELEX cycles due to low resolution and separation efficiency. Moreover, eluting nucleic acids tightly bound to target ligands is challenging. The nucleotide affinity to their targets can be influenced by selection conditions. In some cases, binding and washing conditions, such as target molecule concentration, buffer composition, time, and volume, are modified in later SELEX cycles to obtain aptamers with high affinity and specificity [29,39].

The nucleotides that bind to the target molecule can be eluted through various methods, including heating, changing the ionic strength or pH, competitive elution by adding an excessive amount of the target substance, or adding denaturing substances such as urea, SDS, or EDTA after the washing step [40].

Only a few oligonucleotide sequences from the initial high-diversity library bind to the target ligand [41]. Therefore, it is necessary to repeat the selection step because these sequences are difficult to separate using common separation techniques due to their low partition coefficient [42]. During repeated cycles of selection and amplification, the diversity of the oligonucleotide sample decreases, resulting in high affinity between the oligonucleotides and the target ligand. This is because oligonucleotides with low affinity do not have the opportunity to interact with the ligand. To quantify oligonucleotides bound to the target molecule during SELEX, researchers often use radioactive markers due to their high sensitivity. However, this method has many disadvantages, including the need for isotope production, high cost, and the risk of health hazards during experiments. An alternative option is to use fluorescent dyes as labelling materials. This method is also sensitive, relatively economical, and simple to handle and analyse [23,43].

During the SELEX process, negative selection is necessary to exclude oligonucleotides that bind non-specifically to the membrane or surface of the beads in the absence of the target ligand [44]. Additionally, oligonucleotides that bind to structurally similar compounds or redundant molecules in the real sample, such as serum albumin proteins, are also removed by negative selection [45]. During negative selection, oligonucleotide samples are incubated with unwanted molecules instead of the target ligand itself. Oligonucleotides bound to unwanted molecules are then eliminated, increasing the specificity of aptamers [46].

Subtractive selection is another method used to improve the selectivity of aptamers against complex target substrates, such as whole cells. This method excludes oligonucleotides bound to uninteresting regions of the complex substrate. In this study, aptamers that were isolated can discriminate between target leukaemia cells and other cells, using a normal human lymphoid cell line as subtractive cells. This technique is suitable for isolating highly selective aptamers against cancer or bacterial cells [47,48].

The selection process is terminated when oligonucleotides bound to the target molecule dominate the oligonucleotide sample or when no significant increase in oligonucleotides bound to the target ligand is observed during two or three consecutive SELEX cycles. Oligonucleotides selected in this way are then amplified with unmodified primers. Subsequently, oligonucleotides are individually selected through cloning and sequencing [49]. Sequence analysis can provide valuable information about the selected oligonucleotides, including the identification of regions with homologous sequences that differ only slightly. These unique sequences are often essential for the aptamer’s ability to bind to the target ligand [50].

Following the selection and identification of aptamers, it is crucial to evaluate the affinity and specificity of each individual aptamer. This accurate assessment of affinity, measured by the dissociation constant (K_d_), and specificity is essential for the aptamer’s subsequent applications. The binding conditions can affect these aptamer characteristics. The SELEX process is continually modified to enhance screening efficiency and enable the screening of aptamers for previously inaccessible ligands using new, efficient separation techniques or novel designs of oligonucleotide libraries [51,52].

In recent years, new, specialised procedures have been implemented in the SELEX process with the objective of improving aptamer selection and its efficiency. Significant advances have been made in the sequencing of DNA and RNA chains of varying lengths, a process known as high-throughput sequencing (HTS). This technology offers the potential to enhance the SELEX process with new advantages. It is practically unachievable to reduce the diversity of a DNA library to a level where it can be sequenced in its entirety by traditional sequencing using the SELEX process. Consequently, potential aptamers are identified by cloning only the resulting enriched library. HTS enables the sequencing of millions of sequences per step of selection, a significant improvement over the original SELEX process, and has the ability to obtain information about the process itself, providing insights into the underlying mechanisms. The acceleration of sequencing facilitates the isolation of a suitable aptamer and its identification. Furthermore, the amount of data obtained by this procedure can be employed to determine the relationship between the structure of the aptamer and its binding ability or function through bioinformatic analysis [53].

The SELEX process offers considerable potential for the utilisation of in silico methods [54], which are based on the knowledge that the specific structure of the aptamer will bind the given target ligand. In order to achieve this, it is necessary to sequence a wide range of structures that are able to bind ligands and to reveal the principles of relationships between the structure of aptamers and their binding capabilities. This bioinformatic approach will facilitate the introduction of the in silico method into the routine practice of aptamer generation, thereby paving the way for their widespread use, due to the efficiency of their tailor-made design.

### 3.2. Cell-SELEX, New Biomarker Discovery

SELEX is typically performed using purified substrates, but whole living cells, parts of cells, and viruses can also be used as substrates [55]. This technology, known as Cell-SELEX, offers several advantages. For instance, the aptamers generated are functional towards the native conformation of the target ligand on living cells. Proteins and/or glycosides are found on the surface of cells or viruses. These may be part of various receptors, receptor kinases, ion channels, proteins mediating intercellular interactions, and virus–host interactions, including agglutinins. These complexes are not easily isolated in their native form. The Cell SELEX approach enables the targeting of these molecules with aptamers in instances where they are not available in purified form or even known to exist in their purest form [56,57]. These targets may be unknown and will be identified for the first time through the use of aptamers. This process allows for the discovery of new targets for drug development [58]. Cell-SELEX is therefore an attractive option for generating aptamers that recognize the native conformation on biological surfaces of localised proteins.

Cell-SELEX is a selection strategy that can be performed without prior knowledge of the cell surface molecules displayed on the target cells. This circumvents the problem of purifying target substrates and enables the de novo generation of cell-specific molecular probes [59]. Consequently, Cell-SELEX appears to have considerable potential for a variety of applications, including the identification of altered cells, including those that have undergone neoplastic transformation. Cell-SELEX represents a methodology that can be employed to develop probes for the assessment of the altered differentiation state of cancer cells that differ from normal cells and for the distinction between different types of cancer. It is anticipated that the development of novel aptamers with high affinity and selectivity for cell surface antigens characteristic of the differentiation status of cells will complement and reduce the cost of phenotyping neoplastic cells currently processed by antibodies [48,60]. The oligonucleotides in this method bind to molecules on the extracellular surface. In Cell-SELEX, the molecular composition of the cell surface is not a crucial factor. Instead, the different cell types used in the selection process are the key parameter, as the resulting aptamers will be useful for the specific recognition of these cells [61]. The surface of the cell membrane is a complex system of molecules. The most significant of these are (glyco-)proteins, glycolipids, and proteoglycans, which collectively form the glycocalyx, a layer of carbohydrates that covers the surface of numerous cell types, particularly those of animal origin. Each of these molecules could be a potential target detectable by Cell-SELEX. At the end of a successful selection, aptamers are generated for multiple different targets. This is significant because any of these molecules can have one or more roles in the development of the cell or in the disease they cause [62]. Cell-SELEX has the potential to generate aptamers for unknown molecules, facilitating new biomarker discovery by purifying the target for the aptamer and subsequently discovering a disease biomarker [63]. Cell-SELEX-generated aptamers may identify novel biomarkers as target ligands that were previously unrecognised as specific cell surface molecules, and this method can be used to de novo identify new biomarkers for a desired cell [64].

Aptamers are designed to bind to molecules on the surface of cells in their native state, preserving all post-translational modifications of the proteins. The cell surface carries a negative charge, which results in repulsion between the DNA polyanion and the cell. Therefore, generating nucleic acid aptamers for cells is challenging. However, the structure-related binding between the aptamer and the target is strong enough to overcome the repulsive force. Furthermore, as the selection process progresses and the procedure becomes more stringent, the contribution of non-specific binding sequences, if any, is minimized. This is in line with the objective of Cell-SELEX [33].

The main steps of Cell-SELEX are similar to those of traditional SELEX, including binding, selection, separation, and amplification [65]. To begin, a library of single-stranded oligonucleotides with a wide range of random sequences is synthesised and then incubated with target cells. Following the washing process, the DNA sequences that have bound to the surface of the target cells are eluted by the denaturation of the DNA aptamers in complexes with the cells at 95 °C. This is then followed by concentration by centrifugation. The resulting set is then incubated with negative control cells (cells that do not express the target biomarker). This process allows for the removal of all ssDNA sequences that bind to the negative control cells. PCR amplifies unbound sequences using a biotinylated reverse primer, enriching specific sequences against the target ligand. Magnetic streptavidin beads capture the biotinylated antisense strand, and NaOH separates unlabelled sense ssDNA. Selection cycles increase the binding affinity of targets to the aptamer, enriching selected sequences. Finally, the enriched sets are sequenced, and representative DNA aptamers are selected for subsequent characterization [33].

In the field of diagnostics and basic research, aptamers that recognise specific types of mammalian cells are promising tools. Cancer cell lines are one of the most widely investigated targets for Cell-SELEX, and many aptamers against cancer cells have been generated to date [66]. For instance, Shangguan et al. [48] reported the creation of a series of DNA aptamers as molecular probes for the study of cancer. The human acute lymphoblastic leukaemia cells were tested using both positive and negative tests. The aptamers obtained demonstrated a high affinity for the target cells.

An additional example of the application of Cell-SELEX is the production of DNA aptamers that selectively bind to virus-infected cells. Viral infection modifies the host cell surface by inserting viral proteins, creating virus-specific targets for designing molecular probes that recognize virus-infected cells. Tang et al. conducted Cell-SELEX against adenocarcinoma epithelial cells infected with the virus. They successfully generated specific DNA aptamers against infected cells by combining it with a negative selection step using uninfected cells. The aptamers are bound to several virus-infected cell lines, indicating that they recognize viral proteins displayed on the host cell surface [67].

Using this approach, the Cell-SELEX protocol was successfully used to generate aptamers that can specifically target cancer cells, which involves positive and negative selection. A negative selection step is necessary to remove sequences that bind to normal cells and to improve the specificity of aptamer candidates [68]. The SELEX process was modified to Cell-SELEX in order to identify aptamers for pathogens such as *Streptococcus*, *Listeria monocytogenes*, *Salmonella*, *Staphylococcus aureus*, *Vibrio parahaemolyticus*, *Escherichia coli*, and *Pseudomonas aeruginosa* [69]. Aptamers were also identified for various cellular targets that are typical for diseases such as synovial inflammation, atherosclerosis, obesity, and hyperglycaemia in diabetes patients [45], as well as for a variety of cancer types—leukaemia, glioblastoma, liver cancer, breast cancer, and human glioma cells [70]. 

### 3.3. SELEX Based on Cellular Uptake

Several aptamers identified through Cell-SELEX have been shown to bind to target cells and have the ability of intracellular transport. For instance, one of the aptamers selected against live *Trypanosoma brucei* cells is rapidly internalised by endocytosis and transported to the lysosome by vesicular transport [71]. Aptamers with this internalisation property can be used for intracellular drug delivery [72].

To produce aptamers that specifically mediate intracellular drug transport for target cells, Wu et al. [73] developed a unique SELEX strategy based on intracellular transport rather than binding. A random library is incubated with the target cells. Unbound variants, as well as variants bound to the cell surface, are removed by washing under stringent conditions (incubation with 0.2 M glycine-HCl, pH 4.0, for 5 min). The incorporated variants are then selectively isolated by cell lysis. The authors identified DNA motifs that can efficiently transfer to human chronic lymphocytic leukaemia B cells using this strategy.

Recently, a similar approach has been used to generate tRNA derivatives that can be incorporated into isolated mitochondria. The resulting RNA motifs can be used to construct a new vector for mitochondrial disease therapies [74].

### 3.4. Tissue-SELEX

One of the most critical aspects of Cell-SELEX is the condition and nature of the cells used for selection. The cells used must always be similar to the physiological conditions in vivo, which is not always easy to achieve. Tissues and organs in a living organism consist of various cell types, which are complex, heterogeneous, and have a morphological structure that is difficult to replicate in cell culture. Therefore, even aptamers selected through Cell-SELEX may fail to recognize targets in vivo. To improve aptamer binding in living complex organisms, tissue selection (Tissue-SELEX) is a favourable choice. Tissue-SELEX uses the entire tissue as a binding target [45]. This technique proved to successfully generate specific aptamers when smooth muscle cell-binding aptamers selected by Cell-SELEX were further selected by Tissue-SELEX in whole arteries in vivo to identify aptamers that bind to arterial walls [75]. Wang et al. [76] identified high-affinity aptamers and target proteins associated with ovarian cancer using tissue sections from patients. They were able to identify and characterise a tumour vasculature-specific sequence from human ovarian tissue.

### 3.5. In Vivo Whole Organism SELEX

In Vivo SELEX is a technique that involves injecting a random library of RNA or DNA aptamers into a living organism, allowing them to circulate throughout the body. This technique is useful for identifying aptamers that can be used for therapeutic purposes. The aptamers bind to specific target tissues while unbound aptamers are eliminated by the kidneys. The target organ or tissue must be removed from the organism for RNA or DNA extraction. Isolated aptamers are then amplified by PCR using specific primers that correspond to the flanking constant regions of the starting library. This ensures that only aptamers accumulated in the target tissue are amplified. A group of new aptamers, obtained from the previous round, are repeatedly injected into the same organism to enrich the bindings with the highest affinity to the target tissue or organ of interest. This process has been described in several studies [46,77,78,79].

If the SELEX procedure is performed using a living animal, the aptamers circulate throughout the organism and can potentially bind to targets expressed in the tissue of interest. This approach does not require prior knowledge of the protein ligands for aptamer selection, making it useful for identifying new tissue-specific markers. In cancer, selecting aptamers that are physiologically representative and bind to tumour tissue is particularly attractive due to the highly complex, heterogeneous, and multifaceted tumour microenvironment. Tissue-SELEX and whole-organism In Vivo SELEX are powerful tools for achieving this [45,80,81].

In Vivo SELEX of the whole organism was also performed on a model of intrahepatic metastases. After 14 rounds of in vivo selection, the researchers identified an aptamer that binds to and blocks the activity of an oncogenic helicase that is upregulated in the tumour [80].

Aptamers have been shown to be suitable for medical applications in many studies, but their implementation remains a challenge.

## 4. Implementation of Aptamers

Aptamers have a multitude of applications in molecular methods, including target binding assays [82], drug delivery [83], diagnostics [84], imaging [85], therapeutics [84], protein purification [86], and biological research. Aptamers therefore have significant potential as a valuable tool for visualising and identifying altered cells or tissues due to the progression of various diseases, including cancer and infectious diseases [87]. In the case of cancer, they possess the ability to effectively identify a wide range of oncological biomarkers, cancer metabolites, and whole cancer cells with high precision and specificity even at low concentrations. Furthermore, aptamers are capable of identifying antigens with low immunogenicity that are challenging to detect by the immune system. This makes the production of monoclonal antibodies against them a challenging endeavour [88]. Aptamers’ ability to detect minor differences between cancerous and non-cancerous cells makes them especially valuable for early cancer detection. Aptamer-based diagnostic methods can be employed to detect a range of molecules associated with cancer, alternatively differentiating cells, molecules that may promote tumour growth, and other cancer biomarkers. This enables a sensitive diagnosis of neoplasia and a more accurate and comprehensive understanding of cancer biology [89].

Despite the advances in computational modelling, the determination of the tertiary structure of aptamers remains a challenge, primarily due to the inherent flexibility of aptamers. The relationship between the aptamer sequence and its function is likely influenced by multiple factors, including not only the primary sequence but also the resulting three-dimensional structure. An understanding of the intricate relationship between sequence and assembly into a tertiary three-dimensional structure enables the design of aptamers with enhanced affinity and specificity. Techniques such as directed evolution and chemical modifications can be employed to modify the aptamer sequence, thereby influencing the 3D folding process and fine-tuning the resulting tertiary structure for optimal target binding [90].

Once the relationship between aptamer structure and function is better understood, aptamers can be widely used in various industries, including early disease diagnosis, biomarker detection and discovery, on-site testing, targeted drug delivery, environmental monitoring, food safety testing, and possibly even gene regulation or tissue engineering and the modulation of protein function. The design of aptamers based on their structure prediction would permit the abandonment of the SELEX process, which acts as a random selection, and the design of aptamers on demand. Furthermore, this approach would allow for the optimisation of their performance in a highly efficient manner [91]. While aptamers have been used in clinical settings to a limited extent, numerous aptamer-based clinical studies are currently underway, with the potential for their wider application in clinical practice [92].

### Mechanisms of Analysis and Detection Strategies

There are three modes of detection when detecting surface-bound target molecules through the specific binding of aptamers to their corresponding ligands:Single-binding mode: Labelled aptamers are used to detect surface-bound target molecules [93].Sandwich mode: Two aptamers, each sensitive to different epitopes of target molecules, are used. Typically, one aptamer is immobilised on the surface of the sensor and binds a ligand to which a second labelled aptamer is then bound. This method involves the use of two aptamers, one for capturing the target ligand and the other for its detection [94].Competitive displacement: One component (aptamer or analyte) is immobilised on the sensor surface. In the case of immobilised aptamers, the labelled target ligand binds to the immobilised aptamer. This standard is competitively displaced from binding by the unlabelled ligand originating from the analyte which reduces the signal intensity of the labelled standard [95].

In addition, due to their oligonucleotide nature and adaptive binding mechanism, aptamers enable new detection strategies:Target-induced structure switching (TISS): This is based on the mechanism of induced binding adaptation. Conformational changes during the binding of a target molecule are used to generate a signal [96].Target-induced dissociation (TID): Because aptamers are oligonucleotides, it is possible to design complementary oligonucleotides that hybridise to the aptamer in the absence of the target molecule. In its presence, the complementary sequence dissociates from the aptamer and is replaced by the target molecule [97].Target-induced regeneration of aptamer fragments (TIR): The aptamer can be divided into two parts that do not interact with each other in the absence of the target molecule. In its presence, aptamer fragments re-join and form a trimolecular complex with the target molecule [98].

## 5. Immunodiagnostic Methods Based on Aptamers

In situations where it is crucial to replicate results, particularly in clinical diagnostics, highly sensitive and specific recognition elements are required. Recombinant antibodies, protein scaffolds, or aptamers appear to be viable alternatives to commonly used antibodies. Although recombinant antibodies are often considered the ideal molecular probe, they have limitations in terms of cost-effectiveness, short half-life, and poor validation. Additionally, batch variability remains a problem. Aptamers, on the other hand, offer a suitable alternative for certain applications due to their known and stable structure, easily reproducible binding sensitivity and specificity, and longer half-life [99,100].

Aptamers are increasingly being employed in a variety of research fields as an alternative to existing methodologies, where their properties may present a superior option to other compounds. Many existing methods can be adapted to utilise an aptamer. Research into new aptamers and their potential applications has been extensive in recent years, with scientists selecting aptamers for a range of applications.

A number of aptamers targeting biomarkers were successfully identified using the SELEX process, for example aptamers against prostate-specific membrane antigen, nucleolin, protein tyrosine kinase 7 [101], epithelial cell adhesion molecule (EpCAM) [102], epidermal growth factor receptor (EGFR) [103], Mucin 1 (MUC1) tumour marker [104], or HER2 positive breast carcinoma [105]. A novel DNA aptamer has recently been isolated and employed to detect SARS-CoV-2 by recognising the spike trimer antigen. The test based on this aptamer demonstrated comparable efficiency to real-time PCR [106]. This study demonstrates the applicability of aptamers in clinical practice, where aptamers can be used to construct a comparably high-quality alternative in certain applications.

A further approach is the use of combined technologies, employing both aptamers and antibodies, for the detection of two analytes in patient samples. An illustrative example is the parallel detection of interleukin-6 with a monoclonal antibody and thrombin with an aptamer. In this instance, the detection molecules (antibody and aptamer) were labelled with green and red quantum dots [107].

The recent discovery of aptamers has enabled the detection of adenine nucleotides, including ATP, in a variety of cells [108,109,110], including cancer cells, where elevated levels are observed compared to healthy cells. This is due to the higher energy demands of cancer cells when growing and proliferating. As ATP is found in increased amounts in the tumour environment extracellularly, elevated levels can promote carcinoma invasion. This suggests that ATP may be a non-specific biomarker for various cancer types and disease progression. Modulating the ATP production to lower levels represents a potential therapeutic approach for cancer. Several methods for the detection of ATP include the use of aptamers. In 1995, an aptamer that binds adenosine and ATP was identified. With few modern modifications to increase its sensitivity, this aptamer has been applied in a number of methodologies with considerable success, with a low LOD of 0.85 pM ATP. The modern approaches led to the construction of an aptamer beacon probe capable of detecting and monitoring ATP levels in cancer cells with fluorometric assay [111]. Another study constructed an aptamer-based chimera that included the ATP aptamer to achieve targeted drug delivery with high efficiency directly into cancerous cells, where a drug was released [112]. This further proves that aptamers might serve as a suitable alternative in various methodologies across scientific fields.

### 5.1. Adoption of Enzyme-Linked Immunosorbent Assay (ELISA) Procedures for Aptamer-Based Analysis

ELISA is a diagnostic method that is widely used in clinical settings. It relies on the specificity of antibodies to detect and quantify target molecules, as well as to determine the concentration of analytes [113]. Despite the considerable success of antibody-based testing, there are several limitations to its use. Such discrepancies may be attributed to the random alterations in the antibodies generated between batches during their production, particularly in the case of polyclonal antibodies. The production of monoclonal antibodies, in particular for low immunogenic molecules, can be a laborious and challenging process. Recently, highly homogeneous antibodies produced by recombinant technologies have been increasingly encountered, which addresses some of the disadvantages. Despite these advances, antibodies remain relatively expensive chemicals. These issues highlight the necessity for an alternative to antibodies in order to enhance the enzyme-linked sorbent assays. Among the various possible solutions, replacing antibodies with a more suitable probe represents an optimal approach. The advent of an aptamer—an alternative molecular recognition element (MRE)—has the potential to supplant or supplement the role of antibodies in ELISA, resulting in an enhanced ELISA, the enzyme-linked aptamer-sorbent assay (ELASA) as illustrated on Figure 3. The term ‘ELASA’ is used in different ways, including enzyme-linked aptamer assay (ELAA) [114], enzyme-linked oligonucleotide assay (ELONA), and aptamer-linked immobilised sorbent assay (ALISA) [7].

Aptamer-based diagnostic methods, such as ELASA, offer several advantages over traditional antibody-based ELISA tests [115]. One such advantage is that aptamers have dissociation constants that can reach as low as the picomolar–femtomolar range, indicating their high binding affinity and sensitivity [116,117]. Moreover, aptamers in ELASA systems can be readily regenerated for repeated use, in contrast to antibodies, which are susceptible to rapid degradation [114].

Zhang et al. [118] developed an aptamer-based sandwich enzyme-linked assay (ELASA) with specific aptamers for the detection of largemouth bass virus. The detection limit of this assay was exceptionally low, and the sensitivity of ELASA was only 13.3% less than that of PCR. However, ELASA was considerably more convenient and less time-consuming. Another diagnostic application of ELASA-based diagnosis is the detection of SARS-CoV-2 [119] and the detection of the NS1 protein of Dengue Virus Serotype 2 [120]. Diáz-Fernandez and Ferapontova [121] discussed the potential of simple and low-cost cellulase-linked ELASA as a prospective diagnostic tool for a rapid and accurate liquid biopsy detection of HER2 and numerous other proteins for which aptamers are available.

Wu et al. [122] discussed the potential for antibodies, which are susceptible to permanent degradation, to be replaced by aptamers in ELASA, which can be easily regenerated for repeated usage. Aptamers can be readily reused following the unbinding of the target by heat, which in turn releases the bound antigens. The aptamer structures can be refolded into a functional configuration by lowering the temperature to ambient levels, and this process can be repeated multiple times without a significant loss of binding affinity or specificity. The results of these studies have confirmed the theoretical assumptions that aptamers can be a suitable substitute for antibodies. Furthermore, their implementation does not require major process changes in the usual methodologies.

### 5.2. Replacement of Immunophenotyping by Analysis Based on Aptamers

In the field of oncology, immunophenotyping is employed to classify tumours and determine their cell lineage or origin. By analysing the expression of specific cell surface markers, clinicians can diagnose different types of leukaemia, lymphoma, and other haematological malignancies, as well as predict prognosis and guide treatment decisions. Aptamers offer a distinctive advantage in flow cytometry applications, as almost any fluorophore can be conjugated to an aptamer with a 1:1 stoichiometric ratio. Aptamers therefore have the potential to revolutionise the field of immunophenotyping by providing highly specific and versatile probes for cell surface marker detection in flow cytometry [123]. Continued research and development in this area is likely to further enhance the utility of aptamers in immunophenotyping applications.

### 5.3. Histochemistry/Cytochemistry Based on Aptamers as an Alternative to Immunohisto-/Cyto-Chemistry

Aptamers have also demonstrated potential in immunohistochemistry (IHC) applications, offering an efficient and sensitive alternative to antibodies in the detection of specific proteins or antigens within tissue samples. In instances where a definitive diagnosis cannot be made solely on the basis of tissue morphology, immunohistochemistry may be employed to ascertain the presence or absence of specific biomarkers, which can guide treatment decisions. The process of performing immunohistochemistry on fixed-tissue sections typically takes between 24 and 48 h to yield a result under optimal conditions [123]. Zheng et al. [124] demonstrated the efficacy of the aptamer probe for immunostaining lymphoma tissues. In comparison to CD30 antibody staining, the CD30 aptamer demonstrated the ability to specifically label classical Hodgkin lymphoma and anaplastic large cell lymphoma cells, while avoiding cross-reactivity with background cells. The temperature was lower (37 °C vs. 96 °C) and the probing times were shorter (20 min vs. 90 min) than those typically required for antibody protocols. Moreover, it demonstrated the absence of non-specific staining or cross-reaction with CD30-negative tissues.

It is worthy of note that there is a certain resemblance between aptahistochemistry (AptaHC) and the established Southwestern histochemistry [125]. Both techniques employ the interaction between nucleic acid probes and targeted proteins to localise cellular proteins in tissue sections. In particular, Southwestern histochemistry exploits the inherent affinity of transcriptional regulatory proteins for their specific nucleotide sequence response elements [123,126].

Zhou et al. [127] were the first to report on aptamer-based IHC for disease diagnosis. The objective of the study was to develop an aptamer-based immunohistochemistry method for the diagnosis of tuberculosis. Mannose-capped lipoarabinomannan is a distinctive surface lipoglycan component that is continuously released from the *Mycobacterium tuberculosis* cell wall, rendering it an optimal biomarker for diagnosis by aptamer detection. The aptamer-based IHC method demonstrated enhanced sensitivity and specificity compared to traditional antibody-based methods for the detection of tuberculosis in tissue samples. Aptamer-based IHC can be employed for differential diagnosis, and when used in conjunction with conventional diagnostic methods, it can significantly enhance diagnostic sensitivity. Furthermore, it can be beneficial in surgical pathology diagnosis.

### 5.4. Lateral Flow Assays

Lateral flow assays (LFAs) are increasingly acknowledged, particularly in areas with limited resources, due to their simplicity, high sensitivity, affordability, rapid testing capability, disposable nature, minimal sample volume needed and suitability for on-site detection [128]. The key to developing a lateral-flow biosensor is to identify biorecognition molecules, design strategies, and labels according to the target [129]. Nanoparticles can be used as labels for qualitative and quantitative detection, and their rigidity can also stabilise the spatial structure of aptamers, ensuring maintenance of their affinity and specificity in complex matrices such as blood or urine [130]. It is possible that the utilisation of an aptamer and nanoparticle combination in LFA may present both challenges and opportunities. In recent years, a plethora of aptamer-based sensors and assays have been developed to detect a diverse range of analytes.

The high specificity of aptamers has already been exploited to develop aptamer-based lateral flow devices (LFDs) that can distinguish different strains of the influenza virus. The device is based on a lateral flow design and employs an antibody and an aptamer, the latter of which is highly specific for a particular influenza strain. The dual recognition element lateral flow assay can not only determine whether a patient has the influenza virus but can also detect a specific virulent strain. This has significant implications for future assays, given that antibodies demonstrate broad ranges of specificity but lack the same degree of exquisite discrimination [131].

Aptamer-based LFAs have been employed to detect a range of proteins, including human epidermal growth factor receptor 2 (HER2), cancer antigen 125 (CA125), C-reactive protein (CRP), osteopontin (OPN), β-conglutin, and vaspin. Antibodies are unable to detect small molecules or toxins. To address this issue, aptamers were developed as molecular recognition probes, offering an alternative to antibodies [132]. The aptamer-based lateral flow assay demonstrates versatility in the detection of a range of analytes, with particularly low detection limits, as evidenced by the LFA for the small progesterone molecule, which exhibited an LOD of 5 nM. This makes aptamers an excellent alternative as a recognition element [89].

## 6. Aptamer Nanoconjugates

In the last decade, nanoparticles (NPs) have enabled the introduction of new signal transduction techniques due to their unique properties. Aptamer-modified NPs have been extensively investigated for medical applications, as they can be used to target cells and distribute drugs [133]. The surface of NPs also allows for further modification. Functional molecules, such as drugs or polymers, can be immobilised on the surface carrying an aptamer. This can control the internalisation of nanoparticles into cells or their function within the cell [134]. By binding the aptamer-modified NPs to the target ligand on the cell surface, the drug can be localised directly in the desired cells or tissues for treatment. Unlike systemic therapy, this method can increase the local concentration of the drug while minimising side effects on healthy tissue [135].

Noble metal nanoparticles with a size of less than 100 nm exhibit unique properties for bioassays, including structural, electronic, optical, and catalytic properties for large-scale and sensitive detection [136]. The majority of aptamer-modified noble metal nanoparticles described in the literature are gold nanoparticles (AuNPs). The immobilisation of DNA on AuNPs was well demonstrated by Alivisatos et al. [137] and Mirkin et al. [138]. The chemical synthesis of AuNPs and their subsequent modification with thiolated DNA has been successfully described for aptamers [139,140]. Aptamer-modified AuNPs can be used in medical applications due to their high biocompatibility. Additionally, AuNPs are particularly suitable for colorimetric analyses [141]. Colloidal spherical gold nanoparticles typically exhibit a red hue, which is heavily influenced by their size, shape, surrounding medium, and interparticle distance. When the distance between the nanoparticles is decreased, plasmon coupling can occur, resulting in a red-violet or even blue colour shift [142]. These distance-dependent optical properties can be utilised in the development of colour-based aptasensors [143]. These sensors rely on the binding of the ligand to aptamer-modified AuNPs, resulting in a colour shift from red (non-agglomerated AuNPs) to violet or blue (agglomerated AuNPs) or vice versa. This change is easily observable either with the naked eye or by monitoring absorption spectra.

Although most AuNPs, except those smaller than 3 nm, do not fluoresce, they can be used in fluorescence assays as a ‘superquencher’ for almost all dyes [144]. This enables an increased sensitivity and efficiency of biosensors based on Förster resonance energy transfer (FRET), a physical phenomenon that describes energy transfer between two fluorophores [136].

Two general applications of aptamer-modified AuNPs in electrochemical sensors have been identified.

AuNPs can be immobilised on the sensor surface to increase the surface area and, thus, the sensor loading, thereby increasing the signal intensity. This application of AuNPs as a ligand in electrochemical sensors is common [145,146].AuNPs can be used as labels to amplify the detection signal [141].

Aptasensors capture their target ligand using several mechanisms, including sandwich mode, competitive displacement, or single-binding mode [147]. When designing biosensors for practical use, it is important to consider the nature of the ligand, the conditions for aptamer binding, and the mechanical limitations of the biosensor. This will ensure optimal performance and accurate detection.

## 7. Aptasensors

The demand for new biosensors is continuously growing due to the importance of the rapid and local detection of low-mass compounds. These compounds include residual antibiotics or drugs, illegal drugs, environmentally toxic substances, chemical weapons, and heavy metals. This detection is crucial for aspects such as population health, environmental monitoring, food safety, and counterterrorism [148,149,150].

The accurate detection of disease-derived metabolites or medically relevant bioactive compounds is crucial in disease diagnosis. Nucleic acid aptamers are a promising alternative for bioreceptors. Denatured aptamers can be regenerated easily and quickly, which is particularly important in biological analyses. Aptamers are also promising due to their high specificity against low-molecular-weight targets, even from structurally similar analogues. The structural flexibility of aptamers also allows for the development of new and unique aptamer-based analytical platforms, such as aptasensors [151].

In the context of diagnostic tests used to identify disease and monitor disease progression and patient response to therapy, it is often necessary to target many different molecules. Aptamer-based biosensors therefore represent an attractive format for this purpose, as they can be developed for different molecules according to the same analysis format. Point-of-care testing technology would help in the development of therapeutic methods, as rapid and accurate diagnostic tests are needed in diseases to enable optimal patient treatment at low cost, so that doctors can quickly adjust drug selection or doses [152,153].

Several biomarkers have shown promise in the early diagnosis of tumours, however, there is still a need for the development of efficient detection methods. In recent times, a significant number of aptasensors have been utilised in clinical samples for the detection of exosomes and circulating tumour cells (CTCs). Exosomes, which are extracellular vesicles released actively from cells, are crucial biomarkers, particularly those originating from tumours. A detachable microfluidic device integrated with an electrochemical aptasensor has been utilised for the sequential examination of cancerous exosomes [154]. Additionally, a fluorescent aptasensor has been created for the evaluation of exosomal tumour-associated proteins [155].

Aptasensors demonstrate excellent theoretical properties in the field of diagnosis of various types of cancer, including breast cancer, prostate cancer, ovarian cancer, and digestive tract cancer. Additionally, they exhibit promising potential in the field of infectious pathogens, such as *Escherichia coli*, *Candida*, dengue virus, influenza viruses, Ebola virus, and Zika virus. In addition to the detection of infectious diseases, aptasensors can also be employed to identify a range of other conditions, including osteoporosis, Alzheimer’s, cardiovascular diseases, hormonal imbalances, and the monitoring of thyroid, liver, sugar, vitamin, and ion levels in patient samples [156].

The number of published works and reports of clinical tests in the field of diagnostics using aptamers remains limited. However, there is a market space for the clinical translation of aptamer-based methodologies for diagnosis. The majority of research investigating the relationship between aptamers and sensors has been conducted on electrochemical aptasensors. The most significant findings in recent years are as follows: A biosensor based on a dimeric DNA aptamer was employed to rapidly detect the wild-type SARS-CoV-2 virus and its alpha and delta variants [17]. A DNA aptamer specific to a biomarker for *M. tuberculosis* was used to detect tuberculous meningitis in patients [157] and a sandwich-type electrochemical aptasensor was developed to detect the *Mycobacterium* antigen [158]. An aptamer-based electrochemical technique for the capture of liver cancer HepG2 cells for the detection of circulating tumour cells (CTCs) in patients with liver cancer has already been established [159]. The screening and monitoring of diabetes mellitus was made possible by measuring glycated albumin with an electrochemical aptasensor [160]. Additionally, cardiac biomarkers have been identified by an electrochemical aptasensor [161]. Aptasensors offer not only theoretical advantages, but their implementation has also yielded results that are not only equal to, but can even show a lower limit of detection (LOD) value than traditionally used methods. For instance, the achieved LOD for the detection of small molecules, namely tetracycline and cocaine, was 5 pM and 100 pM, respectively, for aptasensors compared to 13 pM and 0.49 nM for detection by immunosensors [89]. Despite their theoretical performance, electrochemical aptasensors present significant challenges in clinical practice. In contrast, QCM sensors operate on a simple principle and provide indisputable advantages, including real-time monitoring and the possibility of testing multiple samples with one sensor after its regeneration.

### Quartz Crystal Microbalance (QCM) Aptasensors

QCM aptasensors could provide a bridging technique for the implementation of aptamer technology in praxis when there is a need for fast and reliable screening.

QCM biosensors based on the principle of aptamers as recognition elements are gaining attention in applications where the need for cheap, direct, fast testing is paramount, as well as in testing where it is necessary to monitor the progress over time.

So far, most efforts have been devoted to the development of the QCM immunosensor based on the antibody–antigen interaction. Aptamers offer several advantages that may make them a suitable alternative to antibody-based assays, particularly when the assay design is carefully evaluated [162].

QCM sensors are formed by a thin disk of crystal with metal, most often gold electrodes, which serve as sensing surfaces on both sides of the crystal. They use the piezoelectric effect to detect mass changes at the interface between the crystal surface and the sample [163,164]. The piezoelectric effect is a physical phenomenon whereby a material generates electric charge under mechanical stress and vice versa. The principle involves exciting a quartz crystal (in the AT-cut) with an alternating voltage conducted on the surface by two metal electrodes, usually made of gold [165]. Quartz is piezoelectric, meaning that when an alternating electric field is applied, it undergoes a shearing deformation. This causes the crystal surfaces to move in a parallel but opposite directions, generating acoustic waves that propagate through the material in a direction perpendicular to its surface [166]. The crystal oscillates at a frequency that can be measured by inserting it into an oscillating circuit. The analyte binds to the surface of the sensor, specifically to the electrodes on the crystal, causing a change in oscillation frequency. This change is proportional to the weight on the sensor’s surface (Figure 4) [167]. Piezoelectric biosensors can record affinity interactions without requiring specific reagents. Real-time monitoring of frequency changes can provide valuable information about molecular interactions or reactions occurring on the electrode surface, such as layer formation, oxidation, or decomposition [168]. This device can detect changes in the resonant frequency (Δf) of a quartz crystal caused by changes in thickness or mass per unit area (Δm) when small masses are attached or removed. It is sensitive at the nanogram level [169].

In biological measurements, the decrease in the resonance frequency depending on the weight attached to the surface is conveyed by an equation derived for the liquid environment from the Sauerbrey equation for vacuum measurements
Δf = −[(2 × f_0_^2^)/n × (c_66_ × ρ_q_)^1/2^] × [(Δm/A) + [(ρ_L_η_L_)/(4π × f_0_)]^1/2^](1)
where Δf represents a change in frequency (Hz), f_0_ is the base resonant frequency of quartz crystal, n is the overtone number, c_66_ = 2957 × 10^10^ Nm^–2^ represents a constant dependent on the crystal used, ρ_q_ is the crystal density (2648 g/cm^3^), Δm/A is the weight change, η_L_ is the viscosity of liquid, and ρ_L_ is its density [170,171].

Modern QCM devices are capable of measuring not only the frequency but also the dissipation or motion resistance, which relay information about the total energy loss due to the deformation of the sensing layer or viscoelasticity.

QCM aptasensors can quickly detect biomarkers in a miniaturised low-cost sensing system or provide real-time information on changes in the presence of a target molecule in a sample. Low production costs, shorter generation time, and no batch-to-batch variability are essential aspects of transfer to clinical practice [172]. As was already established, the ability of QCM to accurately detect and quantify mass changes with high sensitivity during real-time monitoring renders it suitable for use in environmental monitoring. QCM has been demonstrated to be applicable in the monitoring of volatile gases in the air [173], and in monitoring the absorption or removal of pollutants from water [174].

Using QCM methodology for real-time monitoring of intermolecular interactions was already proven to work by various scientific groups. The most significant interest is directed towards the monitoring of interactions, which are most often observed between the recognition layer of biological origin immobilised on the surface of the sensor and the target ligand in the sample, which flows through the surface of the crystal in the flow cell. This QCM methodology was used by Osypova et al. [175] and described the detection of a low-molecular-weight amino acid derivate by an aptamer, using QCM, proving its ability to detect very small molecules, supported by the detection of an enzyme lysozyme by Bayramoglu et al. [176].

The identification of proteins can offer significant insights for clinical diagnostic purposes. Thrombin being a biomarker important for diagnosis and prevention, its early detection was achieved by Xi et al. [177] using a gold nanocage QCM aptasensor with a low LOD for buffer and human serum. Iijima et al. [178] and Deng et al. [179] successfully developed two QCM aptasensors with different construction to recognize thrombin. As for a different protein, Yao et al. [180] designed a rapid and sensitive, label-free QCM aptasensor to enable the rapid detection of immunoglobulin E (IgE) in samples of human serum.

The timely identification of pathogenic substances is crucial in preventing the rapid transmission of infectious diseases and guaranteeing prompt medical intervention. QCM aptasensors proved their worth by reliably detecting Sars-Cov-2 [181,182], *Brucella* sp. in milk [183], avian influenza virus [184], and hepatitis B [185] and C [186] viruses, as well as the detection of toxins in the environment [187,188,189]. A number of studies have demonstrated the value of aptamers in QCM methodology, particularly in the field of clinical diagnostics. Nubel et al. developed a QCM aptasensor for the monitoring of interactions between aptamer and tumour necrosis factor-alpha [190]. Deletions in the EGFR gene are frequently associated with lung cancer. A DNA aptamer-based QCM biosensor was developed for the detection of a lung cancer biomarker from liquid biopsy samples. This was achieved through the process of hybridization between an aptamer and a DNA sequence containing specific mutations [191].

In recent times, the application of aptasensors has been concentrated on the detection of whole cells, with the objective of facilitating the diagnosis of diseases such as leukaemia or carcinoma. Aptamers form a monolayer on the quartz crystal surface with numerous binding sites that are compatible with biomarkers on the surface of target cells. The affinity binding response is therefore robust and easily quantifiable, and aptamers can therefore represent a more sensitive alternative to antibodies with simpler handling. A variety of aptamers have already been identified against cells or targets expressed on their surface, with a particular focus on cancer cells. Some of these aptamers have already been combined with the QCM methodology for the detection of CCRF-CEM leukemic cells [192] and HepG2 liver cancer cells [159], with low detection limits achieved. Although there is still need for rigorous clinical testing, the results obtained in the laboratory indicate that aptasensors can be transferred to practice thanks to the simplicity of the QCM method and the stability of the aptamers.

## 8. Conclusions and Future Prospects

A considerable number of works laid the theoretical foundations for the use of aptamers for a multitude of purposes, including the detection of various substances and the clinical treatment of diseases. The question remains as to whether this technology can be successfully implemented in practice. The methodology of quartz crystal microbalance (QCM) aptasensors offers new opportunities for business. Aptamers can compete with antibodies in terms of effectiveness, and in some applications, their use is more advantageous. This is due to their properties, including enhanced stability, synthetic preparation without differences in batches, small size, easy regeneration, and modification. The greatest potential for clinical applications appears to be in the field of diagnostics, where the use of aptamers offers the possibility of developing rapid, reliable, and user-friendly diagnostic methods. QCM aptasensors have been demonstrated to function effectively in the context of human samples and to enable the real-time monitoring of interactions. As the basis of the method is the analysis of a sample passing through a flow cell, this could provide new ways of monitoring disease progression. Conversely, when the necessity arises to analyse a considerable number of samples, aptamers can be employed in immunoassay methods, most notably as an alternative to antibodies in ELISA.

This review outlines the potential of aptamers as recognition elements in the future. To achieve a more widespread utilisation of aptamers in practice, it is necessary to address a number of the limiting factors. A deeper understanding of the molecular mechanisms underlying aptamers’ specificity to their targets will undoubtedly facilitate the wider utilisation of these molecules by determining the relationship between aptamer sequence and its function, which allows for rational design and enhanced performance. Although aptamers are mostly non-immunogenic, optimising in vivo stability in complex biological environments like the human body will widen the possibilities of their use, not only in diagnostics, but also in therapy. Outside clinical practice, there is a great deal of promise in using aptamers for cellular targeting and protein function modulation, which should be the focus of future research.

## Figures and Tables

**Figure 1 ijms-25-06833-f001:**
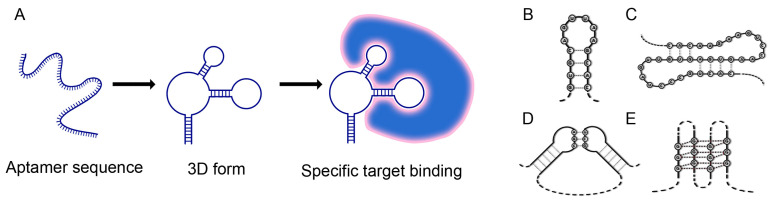
(**A**): Primary sequence of an aptamer is folded into a 3D structure to recognize its target. B to E represent options of 3D structural configurations: (**B**) a hairpin, (**C**) pseudoknot, (**D**) joined hairpins, and (**E**) quadruplex.

**Figure 2 ijms-25-06833-f002:**
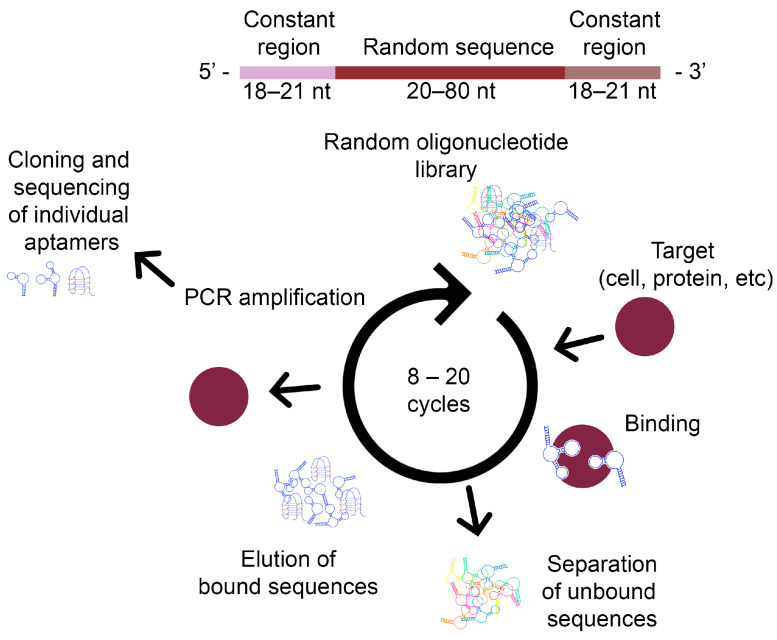
A schematic illustration of the SELEX process for identifying novel target-specific aptamers. This process can be carried out in vitro.

**Figure 3 ijms-25-06833-f003:**
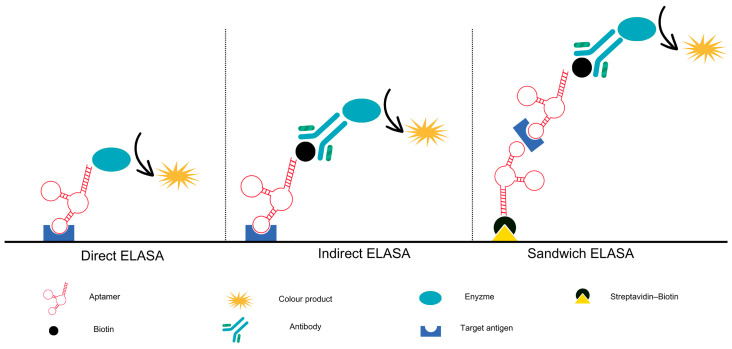
A schematic of the enzyme-linked aptamer-sorbent assay shows different approaches: direct ELASA, indirect ELASA, and sandwich ELASA. In direct ELASA, the aptamer can be directly conjugated to an enzyme or another reporter molecule. In indirect ELASA, the aptamer is first labelled with a reporter molecule such as biotin and then forms a complex with a secondary antibody–enzyme conjugate. The sandwich ELASA involves binding a biotinylated aptamer to capture the target antigen on a surface; subsequently, it uses the same aptamer-reporter molecule and secondary antibody pairing to create a colorimetric change.

**Figure 4 ijms-25-06833-f004:**
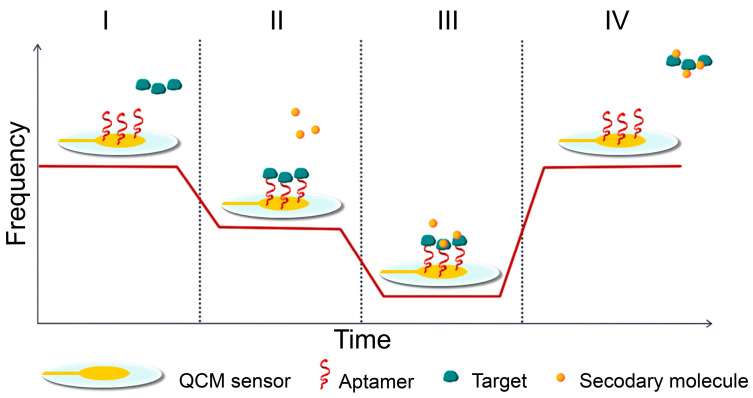
The principle of QCM aptasensor detection is illustrated in the graph, which shows the frequency changes in response to binding events. The graph shows the following stages: I—the initial frequency of the QCM sensor with an immobilised recognition element, which is an aptamer; II—the binding of target molecules; III—the binding of labelled secondary molecules; and IV—the regeneration of the sensor surface.

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
