# Peer review of "New Insights into Aptamers: An Alternative to Antibodies in the Detection of Molecular Biomarkers"

_ijms, 2024, doi:10.3390/ijms25136833_

Round 1

Reviewer 1 Report

Comments and Suggestions for Authors

The paper highlights the transformative potential of aptamers in various diagnostic and therapeutic applications, positioning them as a promising alternative to traditional antibodies. However, the author primarily describes the advantages of aptamers and does not provide much information on their disadvantages compared to traditional antibodies. It would be beneficial to discuss the limitations and potential drawbacks of aptamers in more detail.

1. The abstract is concise and provides a good summary of the paper's content. However, it could benefit from mentioning the challenges and limitations of aptamer technology to provide a more balanced overview.

2. The description of the SELEX process is detailed and informative. However, I suggest discussing recent advancements in technology, such as high-throughput sequencing and in silico methods, to enhance the paper.

3. The section on Cell-SELEX is well-written and provides a good overview of its advantages and applications. Including more specific examples of successful Cell-SELEX experiments, along with their outcomes, would add depth to the discussion.

4. For the applications:

·         The potential for aptamers in emerging fields such as personalized medicine and precision diagnostics could be explored further.

·         The explanation of QCM aptasensors is clear, but more technical details on the design and optimization of QCM sensors could be included for readers interested in developing such sensors.

Comments on the Quality of English Language

the English language is well-written

Reviewer 2 Report

Comments and Suggestions for Authors

This interesting review paper delves into the inherent properties of aptamers in search for novel aptamer designs enabling enhancements in their affinity and specificity toward various biological targets, such as proteins, peptides, glycosides, nucleotides, amino acids, antibiotics, and other potential targets, including small and large organic molecules, and even the whole organic body constructs, like cells and tissues. The Authors explore the novel aptamer designs and the SELEX aptamer generation process to develop new alternative approaches for biorecognition methods, medical diagnostics, and therapeutics. With this fundamental approach and the depth of analyses performed, this paper can serve as a comprehensive resource for researchers and developers of novel diagnostic technologies. This review also covers the immunodiagnostics methods based on aptamers and introduces the ELISA-enzyme-linked apta-sorbent assay (ELASA) that can be applied in several forms depending on the way aptamers is used. The histochemistry/cytochemistry methods based on aptamers are also emphasized. In further analyses, the Authors discuss the utility of quartz crystal microbalance and lateral flow assays, based on aptamers, for the detection of disease biomarkers, drugs, and others.

I recommend the paper for publication after minor revision addressing the issues listed below.

1.   Valuable parameters of aptasensors, such as the Limit of Detection (LOD) and the Limit of Quantification (LOQ), are not discussed. A citation of LOD value for the best aptasensors would be useful as the key parameter for Researchers selecting an analytical method to be applied.

2.            Recently, it has been shown that monitoring and modulation of ATP in cancer cells could be achieved using a rapid fluorometric aptamer probe, see Int. J. Mol. Sci. 2023, 24(11), 9295; Int. J. Mol. Sci. 2021, 22(23), 12940. For the benefit of General Readership, examples of the utilization of aptamer-based arrays to solve some key biological processes of the molecule adenosine triphosphate should be added.

3.              Formatting and English errors:

-       Abstract, lines 19-20: “We explore … exploring”. Better would be “We analyze … by exploring”. Please check.

-       Page 1 line 36: “low … molecules” – change to “small … molecules”.

-       Page 1, line 37: “bud even whole cells and tissues” – replace with: “but can even be applied to the whole cells and tissues”. Please check.

-       Figure 3 Caption: “The sandwich ELISA involves binding…” should be “The sandwich ELASA involves binding...”
